EMBO
Molecular Medicine

# Loss of RASGRP1 in humans impairs T-cell expansion leading to Epstein-Barr virus susceptibility

Sarah Winter[1,2], Emmanuel Martin[1], David Boutboul[1], Christelle Lenoir[1], Sabah Boudjemaa[3],
Arnaud Petit[4], Capucine Picard[1,2,5], Alain Fischer[2,6,7,8], Guy Leverger[4] & Sylvain Latour[1,2,*]

## Abstract

Inherited CTPS1, CD27, and CD70 deficiencies in humans have revealed key factors of T-lymphocyte expansion, a critical prerequisite for an efficient immunity to Epstein–Barr virus (EBV) infection. RASGRP1 is a T-lymphocyte-specific nucleotide exchange factor known to activate the pathway of MAP kinases (MAPK). A deleterious homozygous mutation in *RASGRP1* leading to the loss RASGRP1 expression was identified in two siblings who both developed a persistent EBV infection leading to Hodgkin lymphoma. RASGRP1-deficient T cells exhibited defective MAPK activation and impaired proliferation that was restored by expression of wild-type RASGRP1. Similar defects were observed in T cells from healthy individuals when RASGRP1 was downregulated. RASGRP1-deficient T cells also exhibited decreased CD27-dependent proliferation toward CD70-expressing EBV-transformed B cells, a crucial pathway required for expansion of antigen-specific T cells during anti-EBV immunity. Furthermore, RASGRP1-deficient T cells failed to upregulate CTPS1, an important enzyme involved in DNA synthesis. These results show that RASGRP1 deficiency leads to susceptibility to EBV infection and demonstrate the key role of RASGRP1 at the crossroad of pathways required for the expansion of activated T lymphocytes.

**Keywords** Hodgkin lymphoma; immunodeficiency; lymphocyte; susceptibility to Epstein–Barr virus; T-cell proliferation
**Subject Categories** Genetics, Gene Therapy & Genetic Disease; Immunology; Microbiology, Virology & Host Pathogen Interaction

## Introduction

During Epstein–Barr virus infection, sustained expansion and differentiation of virus-specific cytotoxic effector T cells are key steps to efficiently control virus-infected cells (Hislop & Taylor, 2015; Taylor *et al*, 2015). Specific-CD8[+] T cells can represent more than 30% of circulating T cells during the primary infection, and this strong expansion is necessary to eliminate proliferating EBV-infected B cells. Several primary immune deficiencies caused by mutations in *SH2D1A, CTPS1, MAGT1, ITK, CD27,* and *CD70* are characterized by a high susceptibility to develop recurrent EBV-driven B-cell lymphoproliferative disorders (LPD), although these patients can also develop other infections (Veillette *et al*, 2013; Cohen, 2015; Tangye *et al*, 2017). The particular importance of T-cell proliferation in anti-EBV immunity is notably exemplified by the CTPS1 and CD70 deficiencies (Martin *et al*, 2014; Abolhassani *et al*, 2017; Izawa *et al*, 2017). CTPS1 is required for the *de novo* synthesis of the CTP nucleotide, a precursor of the metabolism of nucleic acids. In T cells, CTPS1 expression is rapidly upregulated in response to TCR stimulation. In the absence of CTPS1, the capacity of activated T cells to proliferate is impaired. Recently, we and others identified a CD70 deficiency in several patients suffering from non-malignant EBV-driven B-cell lymphoproliferative proliferations and EBV-positive Hodgkin lymphoma (Abolhassani *et al*, 2017; Izawa *et al*, 2017). We showed that interactions of CD70 with its ligand CD27 play an important role by providing signals required for the expansion EBV-specific T cells (Izawa *et al*, 2017). CD70 is highly expressed on EBV-infected cells and drives proliferation of CD27-expressing T cells. More recently, null homozygous mutations in *RASGRP1* were reported in two patients with combined immunodeficiency associated with pulmonary infections and persistent EBV infection including EBV-driven Hodgkin lymphoma (Salzer *et al*, 2016; Platt *et al*, 2017). *RASGRP1* codes for a diacylglycerol (DAG)-regulated guanidine exchange factor (GEF) preferentially expressed in T and NK cells (Hogquist, 2001; Kortum *et al*, 2013). RASGRP1 is a specific activator of the small G protein RAS that in turn activates the cascade of Raf-MEK-ERK kinases (also termed as the MAP kinases/MAPK cascade). T-cell antigen receptor (TCR)-mediated RAS-to-ERK activation is mainly dependent on RASGRP1 in human primary T cells (Roose *et al*, 2005; Warnecke *et al*, 2012).

1  Laboratory of Lymphocyte Activation and Susceptibility to EBV infection, Inserm UMR 1163, Paris, France
2  Imagine Institut, University Paris Descartes Sorbonne Paris Cité, Paris, France
3  Department of Pathology, Armand Trousseau Hospital, Paris, France
4  Department of Hematology and Pediatric Oncology, Armand Trousseau Hospital, Paris, France
5  Centre d'Etude des Déficits Immunitaires, Necker-Enfants Malades Hospital, AP-HP, Paris, France
6  Department of Pediatric Immunology, Hematology and Rheumatology, Necker-Enfants Malades Hospital, Assistance Publique-Hôpitaux de Paris (APHP), Paris, France
7  Collège de France, Paris, France
8  Inserm UMR 1163, Paris, France
    *Corresponding author. Tel: +33 1 42 75 43 03; Fax: +33 1 42 75 42 21; E-mail: sylvain.latour@inserm.fr

Rasgrp1-deficient null mice have been reported and exhibited a marked deficiency in development of mature thymocytes and lymphocytes, that was associated with a lack of proliferation in response to TCR stimulation (Dower *et al*, 2000; Hogquist, 2001; Priatel *et al*, 2002). Lymphocytes of the first patient described with RASGRP1 deficiency exhibited several defects, including impaired MAPK/ERK activation, proliferation, cytotoxicity, and migration (Salzer *et al*, 2016). Interestingly, these observations revealed unexpected roles of RASGRP1 in cytoskeletal dynamics during exocytosis of lytic granules in NK and T cells and T-cell migration by its ability to interact with the dynein light chain DYNLL1 and to activate RhoA, respectively. However, the link between these different defects and the susceptibility to EBV infection was not addressed in this study.

In the present work, we report two siblings both presenting recurrent EBV infection caused by a RASGRP1 homozygous loss-of-function mutation and analyze the role of RASGRP1 in the immune response to EBV. We demonstrate that RASGRP1 is essential for T-cell proliferation pathways required for an efficient immune response to EBV, thus providing a relevant explanation for the susceptibility to EBV in humans with RASGRP1 deficiency.

## Results

### Identification of a deleterious mutation in RASGRP1 in two patients with EBV-driven Hodgkin lymphoma

We studied two siblings of a single consanguineous family (Fig 1A). Patients suffered from an exquisite susceptibility to EBV infection and Hodgkin lymphoma. The index case (P1.1) developed mixed-cellularity EBV-positive Hodgkin lymphoma at the age of 5 years, and he was treated by chemotherapy and autologous hematopoietic stem cell transplantation. He then had several episodes of EBV-triggered lymphoproliferation that were sensitive to anti-CD20 (rituximab) administration (Fig 1B). His sister (P1.2) developed at the age of 6 years a scleronodular EBV-positive Hodgkin lymphoma, and she was treated by chemotherapy. She had also an adrenal EBV smooth muscle tumor at the age of 7 years requiring surgery. She died at 11 years of age following relapse of Hodgkin lymphoma. While they were not receiving any immunosuppressive treatment, both patients also presented pneumonia, and disseminated tuberculosis for P1.1 and *Pneumocystis jirovecii* pneumonia for P1.2, respectively. Immunological investigations in P1.1 and P1.2 were carried out 3 and 4 years after chemotherapy, respectively. They revealed significant abnormalities including lymphocytopenia notably characterized by decreased counts of B cells, naïve $CD4^+$ and $CD8^+$ T cells, NK cells, MAIT and absence of iNKT cells, and impaired T-cell proliferation in response to PHA, OKT3, and *Tetanus toxoid*. The lymphocytopenia in P1.2 was more severe than in P1.1 possibly because of the lymphoma relapse at the time of the analysis. Serum immunoglobulin levels were normal or slightly increased (Table 1). These observations strongly suggested that the immunodeficiency in the two patients mostly resulted from a T-cell immunodeficiency.

We performed whole-exome sequencing (WES) that identified 20 homozygous variations in patient P1.1. Only one of them appeared to be deleterious and was not found in public databases

ExAc, 1000 genomes and in the database of our institute containing 10,432 exomes. The mutation corresponds to a two nucleotides insertion in the exon 16 of the *RASGRP1* gene (c.1910_1911insAG) leading to a frameshift that resulted in a premature stop codon p.Ala638GlyfsX16 (or A638GfsX16) (Fig 1C). The mutation was then verified by Sanger sequencing in the family (Fig 1D). Both patients were homozygous for the mutation, while parents were heterozygous and healthy siblings were heterozygous or wild-type carriers, confirming the autosomal recessive inheritance mode of the mutation (Fig 1D). The premature stop is predicted to remove the last 160 amino acids of C-terminal part of RASGRP1. *RASGRP1* transcripts in cells of patient P1.1 were examined by PCR and found to be expressed to a level comparable to that of control cells (Fig 1E). Cloning and sequencing of the PCR products from patient's cells revealed that the majority corresponded to aberrant out-of-frame transcripts lacking at least exon 16 (data not shown). One clone out of 12 was found to correspond to the *RASGRP1* c.1910_1911insAG transcript (data not shown). In contrast, all PCR products from control cells corresponded to the *RASGRP1* wild-type transcript. Further analysis of RASGRP1 protein expression showed no detectable RASGRP1 protein in the lysates from patient' T cells, while in contrast, RASGRP1 was readily detected in cell lysates from healthy donors, migrating as two species that likely differ by post-translational modifications (Fig 1F). The epitope in RASGRP1 recognized by the anti-RASGRP1 antibody used in these experiments was not known. However, this antibody was able to recognize C-terminal truncated RASGRP1 proteins in lysates from HEK293 cells transiently transfected with the cDNA *RASGRP1* c.1910_1911insAG coding RASGRP1$^{Ala638GlyfsX16}$, as well as full-length RASGRP1 in lysates from HEK293 cells transfected with a cDNA coding wild-type RASGRP1 (Fig 1F, right panel). The two other anti-RASGRP1 antibody commercially available did not allow detection of endogenous RASGRP1 from T-cell lysates (Fig EV1). Taken together, these data indicate that the mutation in *RASGRP1* identified in the two patients is deleterious leading to the lack of RASGRP1 expression. The clinical phenotypes as well as the immunological parameters in the two patients reported here are very similar to those initially described in one patient with a RASGRP1 deficiency (Salzer *et al*, 2016) and more recently in another patient (Platt *et al*, 2017). Therefore, these results strongly support that RASGRP1 deficiency is associated with a specific susceptibility to EBV infection.

### Defective ERK1/2 in activated RASGRP1-deficient T cells

Because RASGRP1 is required for T-cell antigen receptor (TCR)-mediated activation of the RAS-to-ERK pathway (Dower *et al*, 2000; Priatel *et al*, 2002), we examined TCR-dependent signals in T-blasts from patient P1.1 and one healthy control upon TCR ligation by anti-CD3 antibody (Fig 2). In patient cells, global tyrosine phosphorylation of substrates of the TCR signaling cascade was not different from that observed in control cells (Fig 2A), albeit $Ca^{2+}$ mobilization and basal phosphorylation of PLC-$\gamma$1 were found to be slightly enhanced (Fig 2B and upper lane of Fig 2C). As expected, phosphorylation of ERK1/2 kinases was found to be markedly reduced in RASGRP1-deficient T cells, when compared to control cells (Fig 2C). In comparison, p38 mitogen-activated kinase and AKT kinase, which are known to be independent of RAS for their activation,

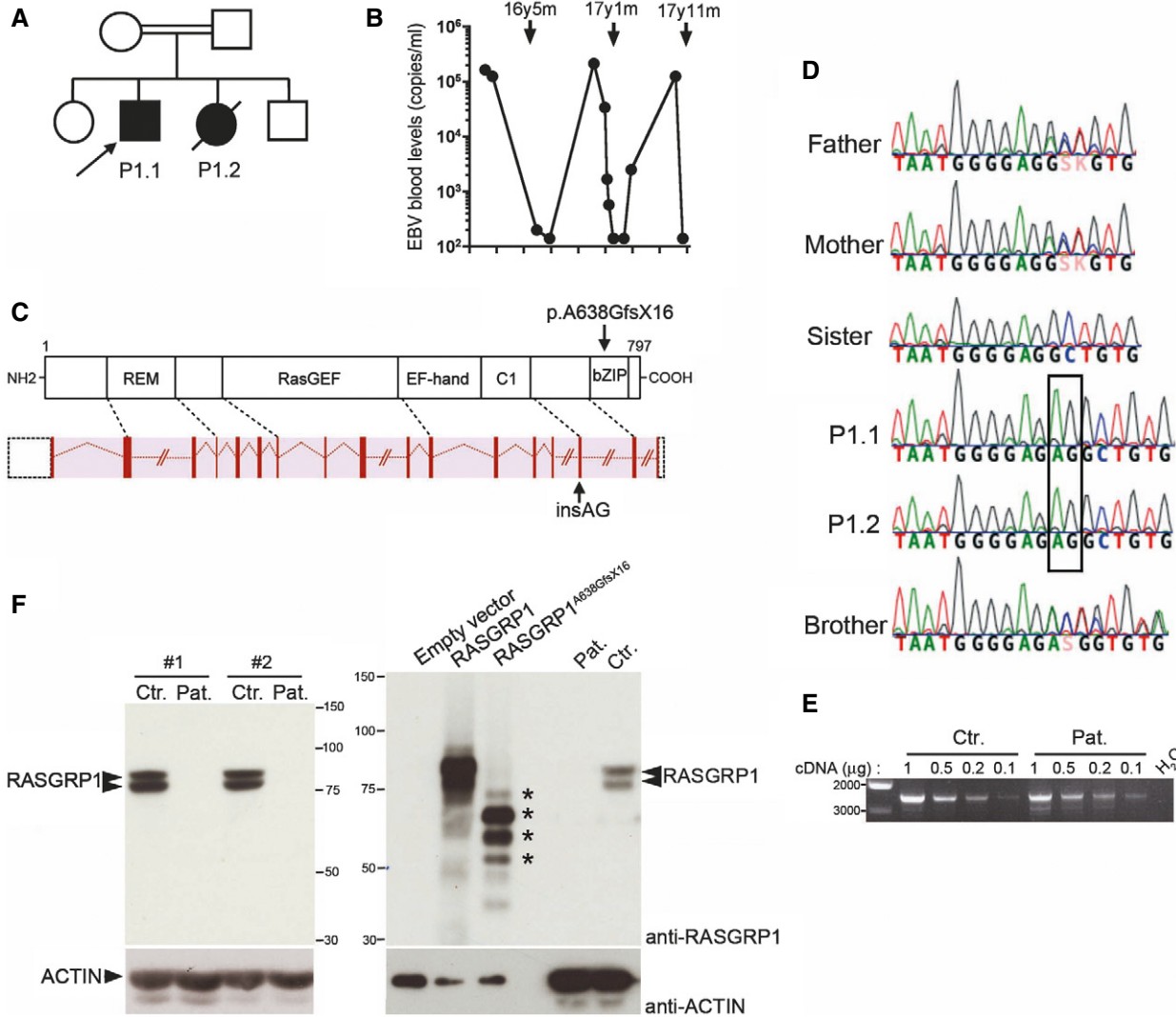

**Figure 1.   Identification of a homozygous loss-of-function mutation in *RASGRP1* in two siblings with Hodgkin lymphoma and defective immunity to EBV.**

A   Pedigree of the family in which the c.1910_1911insAG mutation in *RASGRP1* was identified. The arrow indicates the proband (P1.1) who was analyzed by WES.

B   EBV load in the blood of patient P1.1 is shown as the number of EBV copies detected by PCR at different time points (black circles). Arrows correspond to the anti-CD20/rituximab treatments received by the patient with the age (year, y; month, m) of patient at the time of the treatment.

C   Schematic representation of intron–exon organization of the *RASGRP1* gene and its correspondence at protein level with the different domains of RASGRP1 shown: the Ras exchanger motif (REM), the Ras-guanine exchange factor (RasGEF), the EF-hand, the C1, and the bZIP domains. The mutation is indicated by an arrow at gene and protein levels.

D   DNA electropherograms of the family showing the g.38786931_38786932insAG mutation in P1.1 and P1.2 that is shown in the box.

E   Expression of RASGRP1 transcript in T-cell blasts of healthy control and the patient P1.1 (Pat.). The relative expression of full-length RASGRP1 transcript was examined by qRT–PCR in T-cell blasts of a healthy control and P1.1. Fourfold serial dilutions of cDNAs (1, 0.5, 0.25, and 0.12) were used for amplification of each transcript after quantitation. Base pair markers are shown on the left. PCR products were verified by sequencing showing the expression of c.1910_1911insAG *RASGRP1* transcript in the cells of the patient.

F   Immunoblots for RASGRP1 expression in T-cell blasts from a healthy control (Ctr.) and P1.1 (Pat.) from two different samples (#1 and #2) (left panel). Comparison of RASGRP1 expression in T-cell blasts of control (Ctr.) and patient (Pat.) and in HEK293T cells transfected with empty vector, WT-RASGRP1 or RASGRP1^A638GfsX16 (right panel). RASGRP1 detection using the anti-RASGRP1 antibody MABS146. Actin was used as a loading control. The presence of truncated RASGRP1^A638GfsX16 species detected in HEK293T is indicated by asterisks in the right panel. One representative of three independent experiments from different blood samples.

Source data are available online for this figure.

were similarly phosphorylated in control T cells and in T cells of P1.1. These data show that the pA638GfsStop16 mutation in RASGRP1 leads to a loss-of-protein expression associated with a defective activation of the RAS-to-ERK pathway in response to TCR stimulation.

**Defective proliferation of activated RASGRP1-deficient T cells**

Rasgrp1-deficient null mice are characterized by impaired proliferation of thymocytes and mature T cells in response to TCR stimulation (Dower *et al*, 2000; Hogquist, 2001; Priatel *et al*, 2002).

**Table 1.  Immunological data of the patients P1.1 and P1.2.**

| | Age-matched normal values | P1.1 (11 years/16 years) | P1.2 (10 years) |
|---|---|---|---|
| Leukocytes (cells mm$^{-3}$) | 4,400–15,500 | /4,500 | 6,600 |
| Neutrophils (cells mm$^{-3}$) | 1,800–8,000 | /2,760 | 5,214 |
| Monocytes (cells mm$^{-3}$) | 200–1,000 | /230 | 190 |
| Lymphocytes (cells mm$^{-3}$) | 1,900–3,700 | 1,200/1,400 | 700 |
| T cells | | | |
| CD3$^+$ (cells mm$^{-3}$) | 1,200–2,600 | **864/1,036** | **602** |
| CD4$^+$ (cells mm$^{-3}$) | 650–1,500 | **300/462** | **238** |
| CD8$^+$ (cells mm$^{-3}$) | 370–1,100 | 336/560 | 322 |
| CD4/CD8 ratio | 0.9–2.6 | 0.9/0.8 | 0.7 |
| TCRγ/δ (%) | 0.2–14 | /1.8 | NA |
| CD31$^+$CD45RA$^+$/CD4$^+$ (recent naïve thymic emigrant) (%) | 43–55 | **/9** | **28** |
| CD45RO$^+$/CD4$^+$ (memory) (%) | 13/30 | **/73** | **64** |
| CCR7$^+$CD45RA$^+$/CD8$^+$ (naïve) (%) | 52/68 | **/12** | NA |
| CCR7$^+$CD45RA$^-$/CD8$^+$ (central memory) (%) | 2–4 | **/3** | NA |
| CCR7$^-$CD27$^-$CD45RA$^-$/CD8$^+$ (effector memory) (%) | 11/20 | **/35** | NA |
| CCR7$^+$CD27$^-$CD45RA$^+$/CD8$^+$ (exhausted effector memory EMRA) (%) | 1–18 | **/50** | NA |
| Vα7$^+$CD161$^+$/CD3$^+$ (MAIT) (%) | 1–8 | **/0.078** | NA |
| Vα24$^+$Vβ11$^+$CD161$^+$/CD3$^+$ (iNKT) (%) | > 0.02 | **/0.001** | NA |
| T-cell proliferation (cpm 10$^{-3}$) | | | |
| PHA (6.25 mg ml$^{-1}$) | > 50 | **/29** | **2.8** |
| OKT3 (50 ng ml$^{-1}$) | > 30 | **/3.2** | NA |
| *Tetanus toxoid* | > 10 | NA | **1.7** |
| NK cells | | | |
| CD16$^+$CD56$^+$ (cells mm$^{-3}$) | 100–480 | 108/112 | **49** |
| CD16$^+$CD56$^+$ (%) | 4–17 | 9/8 | 7 |
| B cells | | | |
| CD19$^+$ (cells mm$^{-3}$) | 270–860 | **192/182** | **56** |
| CD19$^+$ (%) | 13–27 | 16/13 | **8** |
| CD21$^+$CD27$^+$/CD19$^+$ (memory) (%) | > 10 | /23 | 25 |
| IgD$^+$IgM$^+$/CD19$^+$CD21$^+$CD27$^+$ (marginal zone) (%) | 31–51 | /1.5 | NA |
| IgD$^-$IgM$^-$/CD19$^+$ CD21$^+$CD27$^+$ (switched) (%) | 21–49 | /19 | NA |
| Immunoglobulin levels (g l$^{-1}$) | | | |
| IgG | 6.6–12.8 | 14.5 | 11.3 |
| IgM | 0.5–2.1 | 1.4 | 0.7 |
| IgA | 0.7–3.4 | 0.6 | 1.72 |

Different immunological parameters of P1.1 and P1.2 were tested from blood and PBMCs: numbers of blood cell populations, different T-cell subsets, B-cell subsets, and natural killer cells from PBMCs (tested by flow cytometry), T-cell proliferation in response to different stimuli (evaluated by incorporation of $^3$H thymidine) and serum (immunoglobulin subclasses). Values in bold correspond to values below the age-matched normal values. MAIT, mucosal invariant cells.

Based on these findings and the known importance of the RAS pathway in cell proliferation, we carefully analyzed the proliferative capacity of T cells from P1.1 (Fig 3). When stimulated with an anti-CD3 or anti-CD3 plus anti-CD28 antibody, T cells of P1.1 weakly proliferated and failed to upregulate the activation marker CD25 as compared to control T cells that strongly divided and expressed CD25 (Fig 3A). This proliferation defect was associated with a block in cell cycle progression with most of the cells failing to enter into the S phase and a strong proportion being blocked in the G2 phase (Fig 3B). Of note, we noticed that after 12–15 days of culture PHA-stimulated RASGRP1-deficient blasts expressing the senescent marker CD57 rapidly accumulated in the culture (Fig EV2). The requirement of RASGRP1 in TCR-dependent proliferation was further confirmed by downregulation of RASGRP1 expression in CD3-activated T cells from healthy individuals by lentiviral transduction of two distinct short hairpin

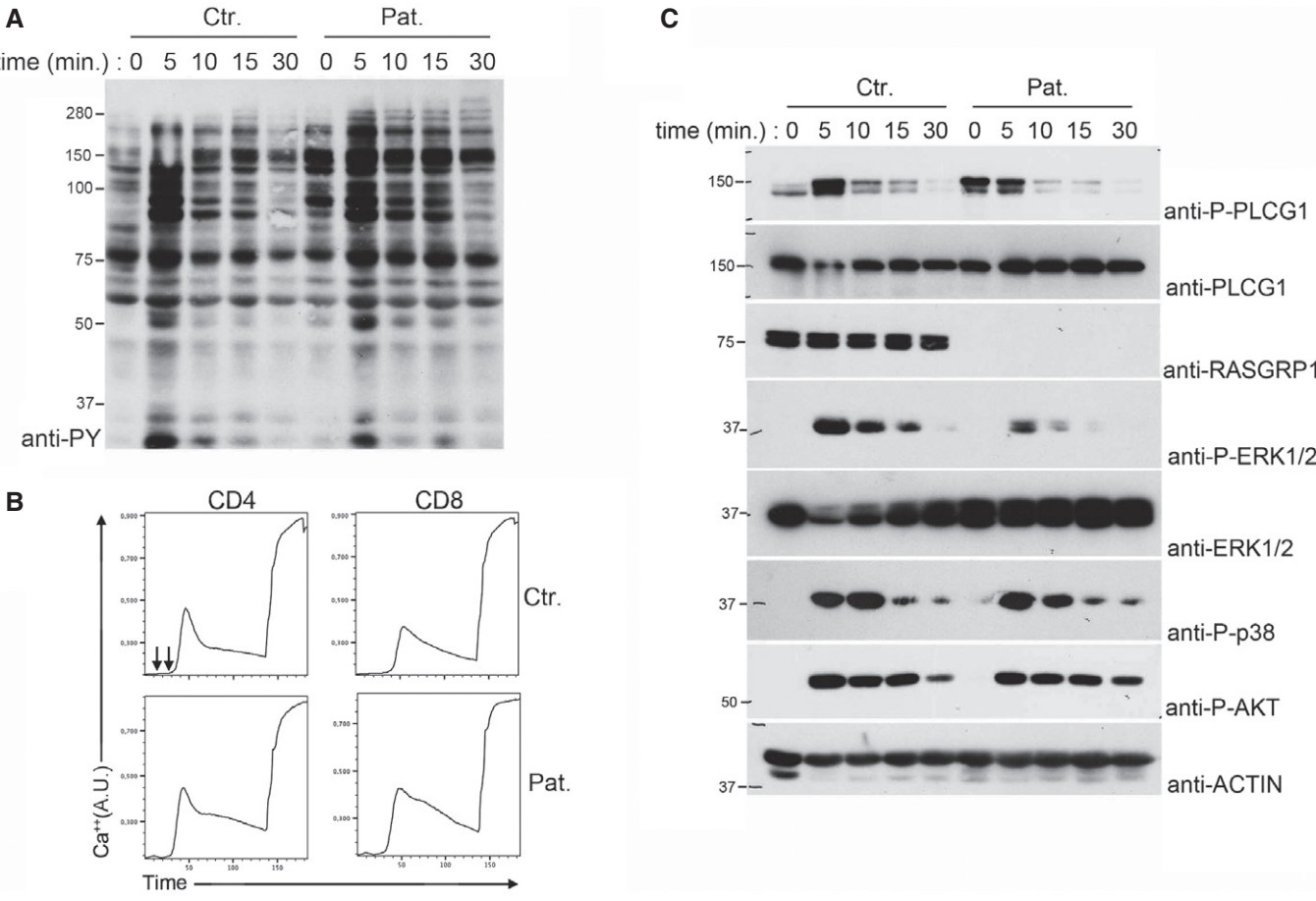

**Figure 2.  Defective ERK1/2 phosphorylation but normal tyrosine phosphorylation signals and calcium flux in activated RASGRP1-deficient T cells.**

A  Immunoblot showing the global tyrosine phosphorylation in T-cell blasts from a control donor (Ctr.) and P1.1 (Pat.) stimulated with anti-CD3 antibodies for 0, 5, 10, 15, and 30 min. One representative of two independent experiments from different blood samples.

B  Flow cytometry analyses of $Ca^{2+}$-flux in T-cell blasts of a control donor and P1.1 loaded with the $Ca^{2+}$-sensitive fluorescent dye Indo-1. Cells were then stimulated with anti-CD3 antibody (first arrow) crosslinked with rabbit anti-mouse antibody (second arrow) and then incubated with ionomycin. Intracellular $Ca^{2+}$ levels are expressed in arbitrary units (A.U).

C  Immunoblots showing phosphorylation of PLCG1 (P-PLCG1), ERK (P-ERK 1/2), p38 (P-p38), and AKT (P-AKT) in T-cell blasts from a control donor and P1.1 stimulated with anti-CD3 antibody for 0, 5, 10, 15, and 30 min. Total ERK 1/2, RASGRP1, and actin (as loading control) are also shown. One representative of three independent experiments from different blood samples.

Source data are available online for this figure.

RNA (shRNA) together with a GFP reporter gene. As control, cells were also infected with an shRNA for CTPS1 previously shown to inhibit CD3-dependent T-cell proliferation (Martin *et al*, 2014). These two shRNAs induced inhibition of RASGRP1 protein expression by 90% (shRASGRP1#1) and 60% (shRASGRP1#2), compared to RASGRP1 expression in cells infected with a lentiviral construct containing a scramble shRNA, a shRNA for CTPS1, or non-infected cells (Fig 3C). Downregulation of RASGRP1 or CTPS1 expression resulted in a growth disadvantage of targeted GFP-positive cells, which rapidly decreased in the culture after CD3 plus CD28 stimulation (Fig 3D). In contrast, the proportion of GFP-positive cells targeted with a scramble shRNA remained stable over time in culture. The rapid diminution of GFP-positive targeted cells with shRNA for RASGRP1 and CTPS1 correlated with their decreased capacity to proliferate in response to CD3

stimulation when compared to cells targeted with a scramble shRNA (Fig 3E).

Because CD70 on EBV-infected B cells is a key driver of immunity to EBV by its ability to trigger CD27-dependent proliferation and expansion of EBV-specific T cells (Izawa *et al*, 2017), we next tested the ability of RASGRP1-deficient T cells to proliferate in response to CD70 stimulation. To this end, T cells were co-cultured with CD70-expressing or CD70-negative EBV-derived B-cell lines (LCLs) pre-incubated with anti-CD3 antibody as previously described (Izawa *et al*, 2017). When co-cultured with CD70-expressing B cells, only a weak proportion of RASGRP1-deficient T cells from PBMCs or T-cell blasts proliferated in contrast to control T cells (Fig 4A and data not shown), while they showed comparable levels of CD27 and CD3 to those expressed by control cells (Fig 4B). These results strongly suggest that RASGRP1-deficient

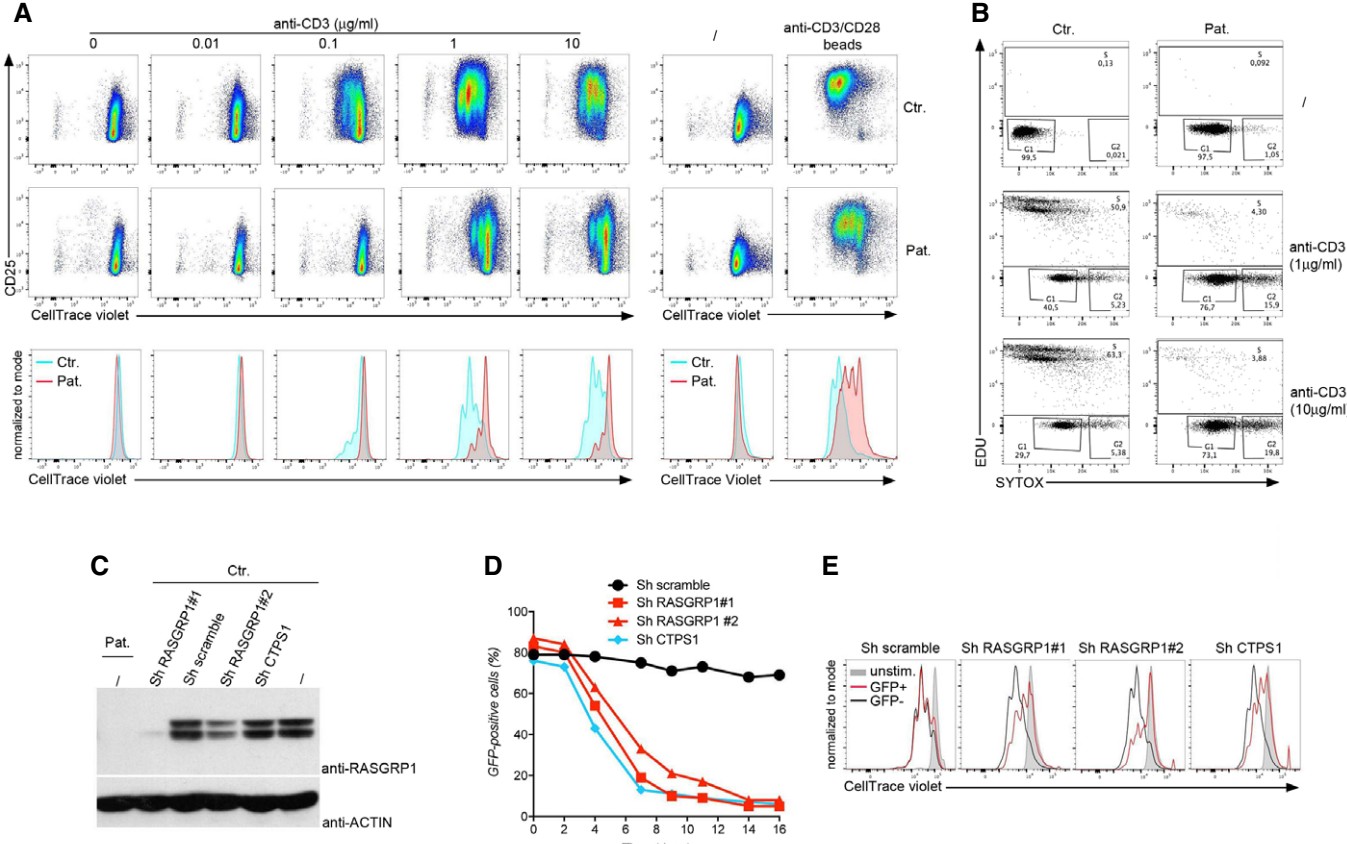

**Figure 3.   T-cell proliferation in response to CD3-TCR activation is defective in RASGRP1-deficient T cells.**

A   Representative dot plots showing cell divisions by dilution of the CellTrace violet dye and expression of CD25 of control donor (Ctr.) or RASGRP1-deficient T-cell blasts gated on CD3$^+$ cells of patient P1.1 (Pat.) stimulated with incremental doses of anti-CD3 antibody or with anti-CD3/CD28-coated beads. Lower panels represent histograms from the dot plots of patient and control that have been overlaid. Each peak of the histograms corresponds to a cell division. One representative of four independent experiments from three different blood samples.

B   Representative dot plots of cell cycle progression of control (Ctr.) and RASGRP1-deficient T cells of patient P1.1 (Pat.) stimulated with two concentrations of anti-CD3 antibody. The percentages of cells in each stage are indicated. Representative data from one of two independent experiments from two different blood samples.

C   Proliferation of T cells in which RASGRP1 expression was silenced with plasmids containing shRNA for RASGRP1 (Sh RASGRP#1 or Sh RASGRP1#2), CTPS1 (Sh CTPS1), or scramble shRNA (Sh scramble) with GFP gene reporter. Immunoblots for RASGRP1 and actin as a loading control of transduced cell cultures with shRNA for RASGRP1 (#1 and #2), CTPS1, or scramble RNA at day 1 after the infection.

D   Curves showing the percentage of GFP-positive cells at different days in long-term expansions after CD3/CD28 stimulation at day 0. One representative of four independent experiments from blood samples of different healthy donors.

E   Representative histograms of CellTrace violet dye dilution showing the cell divisions of GFP-positive (GFP$^+$) and GFP-negative (GFP$^-$) cells after anti-CD3 restimulation with histograms in gray corresponding to unstimulated cells (unstim.). One representative of two independent experiments.

Source data are available online for this figure.

EBV-specific T cells failed to normally expand due to defective proliferation in response to TCR/CD27 activation. Supporting these observations, HLA-A*11-restricted EBV-specific T cells were not detectable in PBMCs from P1.1 patient, while they were present in PBMCs from a control donor, both carriers of HLA-A*11 genotype (Fig 4C).

In contrast, TNF-α and IFN-γ production as well as activation-induced cell death of RASGRP1-deficient T cells in response to CD3 stimulation was not significantly different from those of control T cells (Fig EV3). Exocytosis of cytotoxic granules of RASGRP1-deficient NK cells and CD8$^+$ T cells from P1.1 patient was also found to be normal or slightly diminished in response to CD3 or toward K562 target cells, respectively (Fig EV4). Our observations indicate that

the defect in RASGRP1 preferentially results in impaired proliferation of activated T lymphocytes.

## RASGRP1 is required for proliferation of T cells in response to TCR-CD3 activation

To formally prove that RASGRP1 deficiency was causing defective T-cell proliferation, complementation experiments were undertaken by lentiviral transduction of wild-type RASGRP1 with a mCherry reporter gene of T cells from P1.1 patient (Fig 5). Expression of RASGRP1 in RASGRP1-deficient T cells (Fig 5A) rescued their proliferative capacity in response to CD3 activation (Fig 5B) and enabled mCherry-positive cells expressing RASGRP1 to expand and

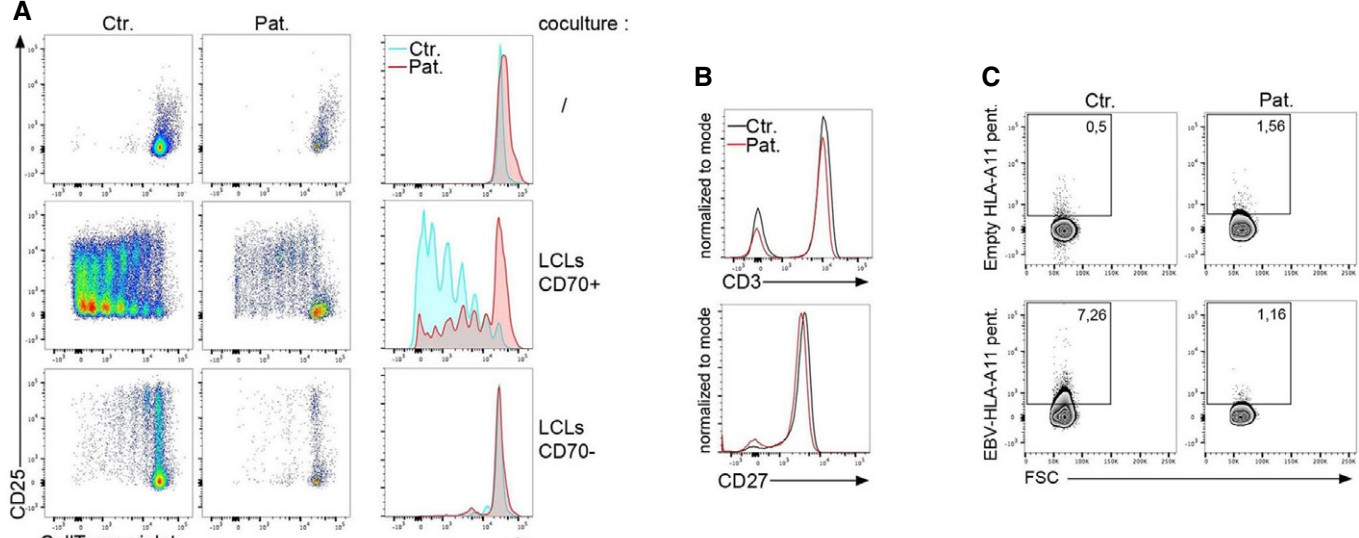

**Figure 4.  CD27/CD70-dependent T-cell proliferation is impaired in RASGRP1-deficient T cells.**

A   Proliferation of T cells from PBMCs from the RASGRP1-deficient patient (Pat.) or healthy donor (Ctr.) cultured during 8 days with irradiated LCLs (+LCLs) expressing CD70 (LCLs CD70+) or not (LCLs CD70−). Irradiated LCLs were pre-incubated with anti-CD3 antibody before to be added to the PBMCs. Representative dot plots of CellTrace violet dye dilution and CD25 expression gated on CD3+ cells and histograms on the right from the dot plots of patient and control that have been overlaid. One representative of two independent experiments from two different blood samples.

B   Histograms of staining for CD3 (upper) or CD27 (lower) expression on PBMCs from a control donor (Ctr.) and the patient P1.1 (Pat.). One representative of three independent experiments from three different blood samples.

C   Dot plots of staining for EBV-specific CD8+ T cells with empty HLA-A11 pentamers or EBV-loaded HLA-A11 pentamers on PBMCs from a HLA-A11 control donor (Ctr.) and the patient P1.1 (Pat.). Numbers correspond to the proportion of positive cells for the staining in the gate.

accumulate in culture when repeatedly stimulated with anti-CD3 or anti-CD3 plus anti-CD28 antibodies (Fig 5C). In contrast, the proliferation rate of mCherry-positive control cells transduced with the RASGRP1-containing vector remained stable with no accumulation of mCherry-positive cells in the culture. Therefore, these results demonstrate the causal relationship between the RASGRP1 deficiency and defective T-cell proliferation in response to TCR activation.

**Defective CTPS1 expression in activated RASGRP1-deficient T cells**

Interestingly, we previously showed that a chemical inhibitor of ERK1/2 kinases inhibited CTPS1 expression in activated T cells, suggesting that the RAS-to-ERK pathway may play an important role in the expression of CTPS1 (Martin *et al*, 2014). Hence, the defective proliferation capacity of activated RASGRP1-deficient T cells may result in part of abnormal CTPS1 upregulation in response to TCR stimulation. We tested this possibility by analyzing CTPS1 expression in T cells from P1.1 following anti-CD3/CD28 stimulation. As previously reported (Martin *et al*, 2014), CTPS1 expression was upregulated after 12 h of stimulation and persisted until 72 h in control cells, whereas in activated RASGRP1-deficient T cells, only a slight and transient upregulation of CTPS1 was detectable after 12 h of stimulation (Figs 6A and EV5). Similarly, GFP-positive T cells in which RASGRP1 expression was inhibited by a shRNA targeting RASGRP1 exhibited a decreased expression of CTPS1 in response to CD3/CD28 stimulation, in comparison with non-targeted

GFP-negative T cells (Fig 6B). In contrast, CTPS1 expression was not modified in cells infected with a scramble shRNA. To verify that the defect of CTPS1 expression was not the consequence of the defective proliferation of activated RASGRP1-deficient T cells, we analyzed CTPS1 expression in normal activated T cells in the presence of deazauridine, an inhibitor of cell proliferation (Martin *et al*, 2014). In these conditions, proliferation of activated T cells was completely blocked (Fig 6C). In cells treated with deazauridine, CTPS1 expression after 48 h of stimulation was found to be comparable to that of cells not treated (Figs 6D and EV5). Therefore, activated RASGRP1-deficient T cells exhibit an impaired ability to upregulate CTPS1 expression suggesting that CTPS1 expression in activated T cells is under the control of the RASGRP1/MAPK signaling. However, we observed that CTP or cytidine addition to the medium was not able to restore TCR-triggered proliferation of RASGRP1-deficient cells (Fig EV5A), in contrast to CTPS1-deficient T cells as we previously reported (Martin *et al*, 2014). Thus, this suggests that the RAS-ERK pathway may exert additional functions required for T-cell proliferation. In particular, other genes known to be involved in proliferation might be controlled by the RAS-ERK pathway. To examine this possibility, we tested the expression of the proliferating cell nuclear antigen PCNA, which plays a central role at the replication fork by recruiting enzymes required for DNA replication (Boehm *et al*, 2016). Like CTPS1, PCNA expression was markedly increased in normal T cells when stimulated by anti-CD3/CD28 antibodies, whereas it was defective in RASGRP1-deficient activated T cells and did not result from the impaired capacity of RASGRP1-deficient cells to proliferate (Fig EV5B), since no effect on PCNA

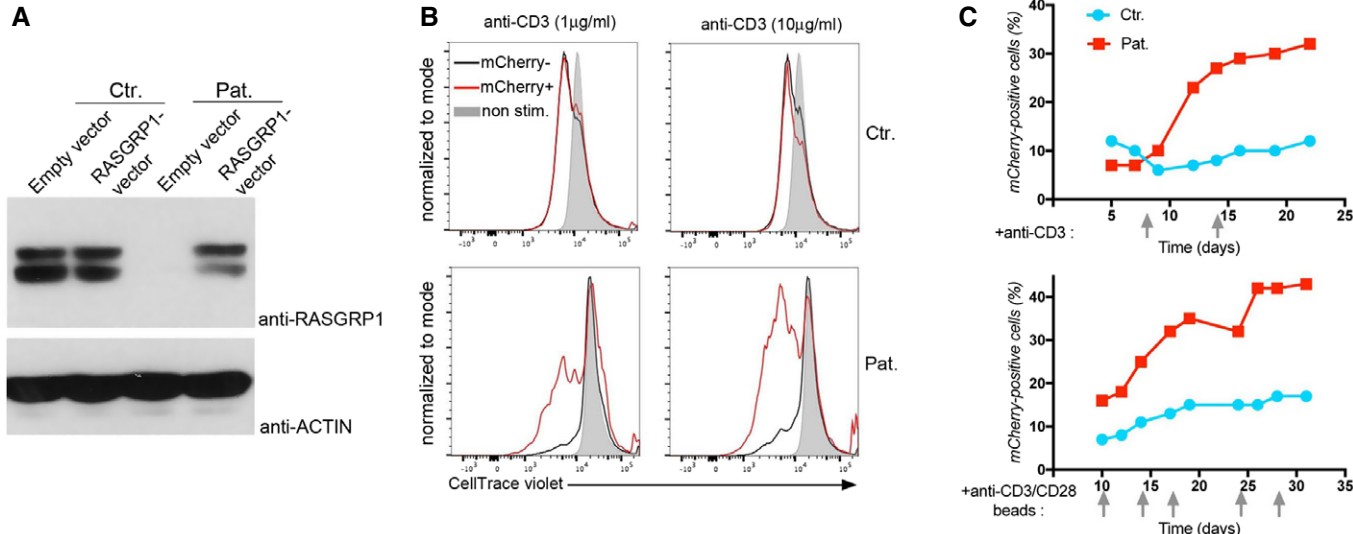

**Figure 5.  RASGRP1 is required for T-cell expansion in response to CD3-TCR activation.**

A   Rescue of T-cell proliferation by expression of wild-type RASGRP1 in RASGRP1-deficient T cells. Immunoblots for RASGRP1 and actin as a loading control of control (Ctr.) and RASGRP1-deficient T cells of patient P1.1 (Pat.) transduced by empty or wild-type RASGRP1-containing vector with mCherry gene reporter at day 14 of culture.

B   Proliferation of control (Ctr.) and RASGRP1-deficient T cells of patient P1.1 (Pat.) transduced by empty or wild-type RASGRP1-containing vector. Representative histograms of CellTrace violet dye dilution showing the cell divisions of mCherry-positive (mCherry⁺) and mCherry-negative (mCherry⁻) cells after stimulation with two concentrations of anti-CD3 antibody, with histograms in gray corresponding to unstimulated cells (unstim.).

C   Curves of the percentage of mCherry-positive transduced cells after repeated stimulations indicated by gray arrows with anti-CD3 (upper panels) or anti-CD3/CD28 (lower panels) antibodies at different days in long-term expansions.

Data information: Representative data from two independent experiments.
Source data are available online for this figure.

## Discussion

expression was noticed in deazauridine-treated activated T cells like it was observed for CTPS1 (Fig EV5C).

By many aspects, the clinical and functional phenotype associated with RASGRP1 deficiency are similar to those associated with CD70, CD27, and CTPS1 deficiencies (Martin *et al*, 2014; Alkhairy *et al*, 2015; Abolhassani *et al*, 2017; Izawa *et al*, 2017). Patients suffering from these genetic diseases exhibit a specific susceptibility to EBV-driven lymphoproliferative disorders, which is very often the most severe phenotype associated with these conditions, even though some patients can develop other infections particularly in CTPS1 deficiency. Like CTPS1, CD70, and CD27, RASGRP1 appears to be critical for expansion of T cells that needs to be particularly intense and sustained during EBV infection (Hislop & Taylor, 2015; Taylor *et al*, 2015). In T cells, CTPS1 expression is rapidly upregulated in response to TCR stimulation and inhibitors of the MAPK pathway strongly diminished the expression of CTPS1 (Martin *et al*, 2014). As activation of the MAPK cascade is mostly dependent of RASGRP1 in T cells (Roose *et al*, 2005; Warnecke *et al*, 2012), the observation that CTPS1 expression is impaired in activated RASGRP1-deficient T cells strongly supports a role of RASGRP1 in CTPS1 gene expression. Interestingly, the proximal promoter of CTPS1 contains binding sites for E2A and MZF1 (Martin E. and S. Latour, unpublished observations), two

transcription factors involved in the suppression of lymphocyte and myeloid proliferation (Gaboli *et al*, 2001; Talora *et al*, 2003). Notably, inhibition of E2A activity is dependent on the MAPK pathway (Bain *et al*, 2001). We showed that *PCNA* gene expression was also defective in RASGRP1-deficient T cells, suggesting that RASGRP1 might more extensively be involved in the expression of genes required for proliferation. Further studies are warranted to understand the role of RASGRP1 in CTPS1 gene expression, possibly through the regulation by E2A and MZF, and to characterize in detail RASGRP1-dependent pathways that control T-cell proliferation.

CD27/CD70-dependent proliferation was also affected in RASGRP1-deficient T cells. We recently demonstrated that CD27/CD70 is a key pathway in immunity to EBV by promoting the expansion of antigen-specific T cells (Izawa *et al*, 2017). Thus, impairment of this pathway in RASGRP1-deficient patients likely accounts for their susceptibility to EBV. However, abnormalities of the exocytosis of cytotoxic granules of NK and CD8⁺ T cells and lymphocyte migration have been reported in the first RASGRP1-deficient patient that could also participate to the susceptibility to EBV (Salzer *et al*, 2016). We analyzed the degranulation of NK and CD8⁺ T cells in one of the two siblings reported here, which was found to be normal in contrast to this previous study. We have no clear explanation for this discrepancy, but it is still possible that the frameshift p.Ala638GlyfsX16 mutation reported herein has a less severe consequence than the mutation previously reported, which corresponds to an early premature stop codon (p.Arg246X). Nevertheless, we

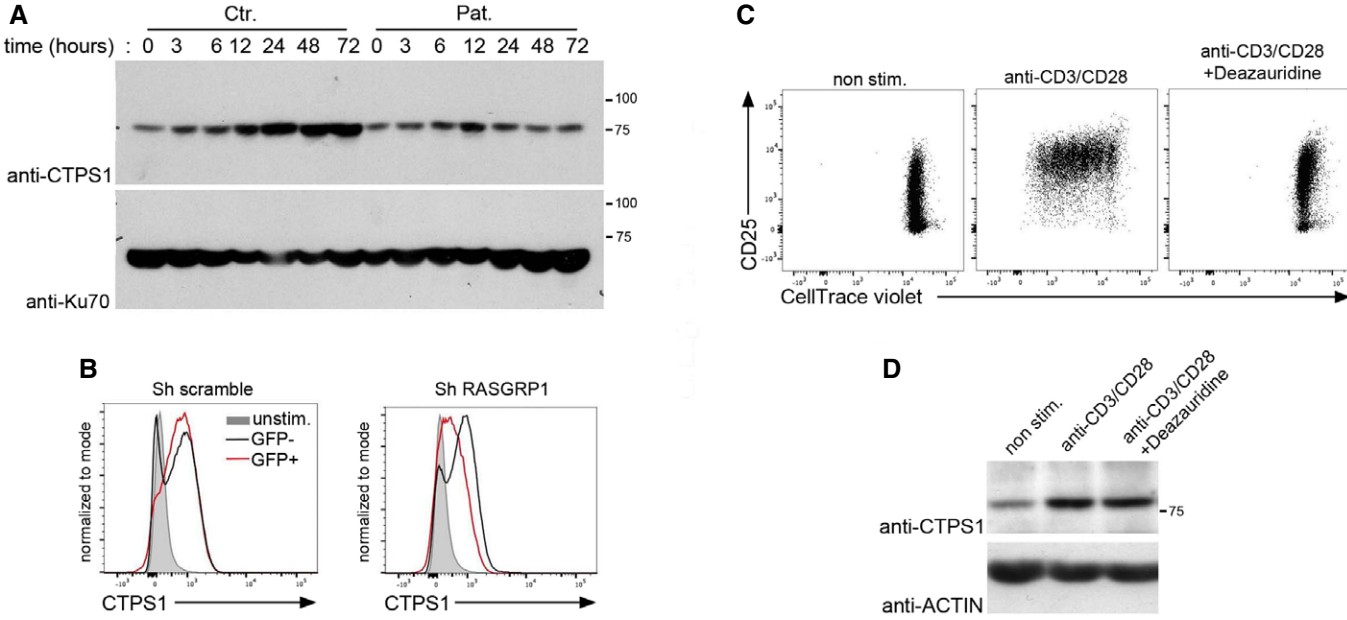

**Figure 6.  CTPS1 upregulation is impaired in activated RASGRP1-deficient T cells.**

A    Immunoblots for CTPS1 expression of control donor (Ctr.) and P1.1 T-cell blasts (Pat.) stimulated with anti-CD3/CD28 antibodies for different periods of time. Ku70 was used as a loading control. One representative of two independent experiments from different blood samples.

B    Histograms of intracellular staining for CTPS1 expression in T cells from a control donor in which RASGRP1 expression was silenced with plasmids containing shRNA for RASGRP1 (Sh RASGRP1) or scramble shRNA (Sh scramble) with GFP gene reporter. Cells were stimulated with anti-CD3/CD28 antibodies or not (non stim.) for 48 h. One representative of two independent experiments from blood samples of two different healthy donors.

C, D   Control donor T-cell blasts stimulated or not (non stim.) with anti-CD3/CD28 beads in the presence or not of 40 μM of deazauridine. In panel (C), proliferation was analyzed at day 3 of stimulation. In panel (D), CTPS1 expression was analyzed at 48 h of stimulation. Actin as a loading control.

Source data are available online for this figure.

failed to detect any expression of RASGRP1$^{Ala638GlyfsX16}$ in cells of the patient. In any case, abnormalities in effector functions may not play a determinant role in the susceptibility to EBV associated with the RASGRP1 deficiency since proliferation and expansion of activated T cells, which precede the effector phase of the immune response are impaired in RASGRP1-deficient patients (Hislop & Taylor, 2015; Taylor *et al*, 2015). However, the two patients also exhibited T-cell and NK-cell lymphopenia with decreased naïve T cells indicative of impaired thymic development leading to decreased production of mature naïve T cells that is often observed in combined immunodeficiencies caused by gene defects affecting T-cell signaling components (Notarangelo, 2014). This is also further supported by studies of mice models showing low numbers of mature thymocytes and T cells in rasgrp1-null-deficient mice and aberrant thymic selection in rasgrp1-deficient mice expressing a C-terminal truncated form of rasgrp1 (Dower *et al*, 2000; Fuller *et al*, 2012). NK cells have not been examined in rasgrp1-deficient mice, but it is likely that RASGRP1 plays a role in NK-cell development or/and homeostasis.

The absence of iNKT cells as found in P1.1 fits with a critical role of RASGRP1 in NKT cell development in mice (Shen *et al*, 2011). This cellular defect might also contribute to the impaired immune response to EBV infection in RASGRP1-deficient patients since iNKT cells have the ability to control EBV-infected B cells and are often defective in primary deficiencies characterized by high susceptibility to EBV (Chung *et al*, 2013; Veillette *et al*, 2013).

RASGRP1 has been also shown to be involved in B-cell development, activation and tolerance both in mice and humans (Bartlett *et al*, 2013; Guo & Rothstein, 2016; Salzer *et al*, 2016). However, both patients reported herein had normal or slightly elevated immunoglobulin levels, although they had decreased B-cell counts. The role of RASGRP1 in B cells could be different or/and less important than in T cells. Along these lines, all RASGRP1-deficient patients have developed EBV-driven B-cell lymphoproliferation, indicating that RASGRP1-deficient B cells retained an intact ability to proliferate upon EBV transformation. Interestingly, *RASGRP1* was identified as a risk locus for autoimmunity (Qu *et al*, 2009; Sun *et al*, 2016) and older *rasgrp1*-deficient mice developed systemic lupus erythematosus-like symptoms (Bartlett *et al*, 2013; Guo & Rothstein, 2016). Autoimmunity was however not noticed in the two patients herein, as well as in the patient previously described (Salzer *et al*, 2016), but all were at young age.

In conclusion, our results demonstrate the critical role of RASGRP1 in TCR- and CD27/CD70-mediated proliferation of T cells, which is known to be critical for an efficient immunity to EBV. Like CD27-, CD70-deficient patients, RASGRP1-deficient patients appear to be particularly prone to develop EBV-driven B-cell lymphoma as the four patients described today had Hodgkin lymphoma. Finally, our observations also support the design of specific inhibitors of RASGRP1 that could provide a potential way to suppress abnormal proliferation of activated T cells occurring in various pathophysiological conditions including autoimmunity.

# Materials and Methods

## Study approval

Informed and written consent was obtained from donors, patients, and families of patients. The study and protocols conform to the 1975 Declaration of Helsinki as well as to local legislation and ethical guidelines from the Comité de Protection des Personnes de l'Ile de France II and the French advisory committee on data processing in medical research.

## Exome sequencing and analysis

Exome capture and analysis were performed as previously described (Martin *et al*, 2014; Izawa *et al*, 2017). The RASGRP1 variation identified in the patient (15:38786930 C/CCT), an homozygous frameshift insertion c.1910_1911insAG p.Ala638GlyfsStop16 was not reported in the exome and genome aggregation consortium (ExAC and gnomAD) databases (http://exac.broadinstitute.org and http://gnomad.broadinstitute.org) nor in available exome public databases (dbSNP, the 1000 Genomes, the NHLBI Exome Sequencing Project) including our institute database containing 10,432 exomes.

## Gene expression analysis and cloning

Total RNA of T-cell blasts from P1.1 and a healthy donor obtained with RNeasy Mini kit (QIAGEN) were used as template for reverse transcription using Superscript II First Strand Synthesis System (Invitrogen). For PCR amplifications and cloning from cDNAs, the following primers were used: Forward: 5′-CACCATGGGCACCCT GGGCAA-3′; Reverse: 5′-GGGCTAAGAACAGTCACCCTGC-3′. PCR products were verified by sequencing showing the expression of RASGRP1 mutated transcript (c.1910_1911insAG) in one clone out of 12 analyzed from P1.1 cells.

## Cell culture

T-cell blasts were obtained from PBMCs and cultured as previously reported (Martin *et al*, 2014; Izawa *et al*, 2017). Before to be tested in the different assays, T-cell blasts were analyzed for CD3, CD4, CD8, CD45RO, CD45RA, and CD57 expression. Phenotypes of T-cell blasts from healthy donors and the patient were comparable for the expression of these different markers before day 12–15 of culture.

## Proliferation and cell cycle

Proliferation of PBMCs or T-cell blasts in response to anti-CD3, anti-CD3/CD28, or in co-culture with irradiated CD70-positive or CD70-negative EBV-transformed LCLs were done following procedures previously reported (Izawa *et al*, 2017).

## Flow cytometry

Cell staining and the flow cytometry-based phenotypic analyses of PBMCs and cells were performed according to standard flow cytometry methods (Martin *et al*, 2014; Izawa *et al*, 2017). The following monoclonal antibodies were conjugated to phycoerythrin-cyanin7 (PE-Cy7) Brilliant Violet 785 (BV785), Brilliant Violet 510 (BV510), Brilliant Violet (BV650), phycoerythrin (PE), phyco-erythrin-cyanin5 (PE-Cy5), Brilliant Violet 451 (BV421), Peridinin-chlorophyll-cyanin5.5 (PerCP-Cy5.5): anti-CD25 (BC96; dilution 1:40), anti-CD3 (OKT3; dilution 1:40), anti-CD4 (OKT4; dilution 1:40), anti CD8 (RPA-T8; dilution 1:40), anti-CD27 (O323; dilution 1:40), anti-CD45RA (HI100; dilution 1:40), anti-CD161 (HP-3G10; dilution 1:20), anti-TCR Vα7.2 (3C10; dilution 1:20) all purchased from Sony Biotechnology Inc.; anti-TCR Vα24 (C15; dilution 1:20), anti-TCR Vβ11 (C21; dilution 1:20), anti-TCR γδ (IMMU510; dilution 1:40) from Beckman Coulter; and anti-CD19 (HIB19; dilution 1:40) and anti-CD57 (NK-1; dilution 1:200) from BD Biosciences. For intracellular staining of CTPS1, cells were fixed and permeabilized using the Intraprep kit (Beckman Coulter) according to the manufacturer's instructions. Cells were stained with an anti-CTPS1 antibody (EPR8086B, Abcam; dilution 1:200) or an isotype-matched antibody and then labeled with a FITC-goat anti-rabbit secondary antibody.

## EBV-specific T-cell detection

HLA genotyping of the patient showed that he was a carrier of HLA-A*01 A*11, B*52, and B*57, for which only HLA-A*11 reagents were available to assess EBV-specific T cells. EBV-specific CD8$^+$ T cells from PBMCs of the patient and a healthy control carrier of HLA-A*11 were detected using a mix of unlabeled EBV HLA-A11:01 Pro5 Pentamers (Proimmune) mixed with R-PE Pro5 Fluorotag in addition with BV785-anti-CD3, APC-anti-CD19, BV510-anti-CD4, and BV650-anti-CD8 antibodies according to the manufacturer's instructions. The EBV HLA-A11:01 Pro5 Pentamers mix contains two different pentamers presenting ATIGTAMYK (residues 134–142 from BRLF1) or SSCSSCPLSK (residues 340–349, from LMP2) peptides derived from BRLF1 or LMP2 proteins of EBV.

## Degranulation

T-cell and NK degranulation was determined by analysis of the expression of CD107/LAMP as previously described (Martin *et al*, 2014) using PE-anti-CD107a (H4A3; dilution 1:200) and PE-CD107b (H4B4; dilution 1:200) purchased from Sony Biotechnology Inc. For NK-cell degranulation, blasts maintained with 1,000 UI ml$^{-1}$ of IL-2 were stimulated in the presence of the K562 target cells at a ratio of 5/1 (blasts/K562) for 3 h and simultaneously labeled with PE-anti-CD107a/b. Cells were then collected, washed, and stained with BV785-anti-CD3 and APC-anti-NKP46 (9E2; dilution 1:40) monoclonal antibody and analyzed by flow cytometry. Activation-induced cell death was examined 12 h after stimulation with coated anti-CD3 antibody by staining with APC-Annexin V Viaprobe (BD; dilution 1:20) and BV421-SYTOX (SYTOX Blue, S34857, Invitrogen; dilution 1:4,000).

## Immunoblotting

Stimulation, extraction, SDS–PAGE, and immunoblotting were performed as previously described (Martin *et al*, 2014; Izawa *et al*, 2017). The following antibodies were used for immunoblotting: antiphosphorylated tyrosine (4G10; dilution 1:1,000), antiphospho-rylated PLC-γ1 (#2821S; dilution 1:1,000), anti-PLC-γ1 (#2822S;

### The paper explained

**Problem**

Primary immunodeficiencies characterized by a high susceptibility to Epstein–Barr virus (EBV) infection are rare genetic disorders. Identification of the molecular causes underlying these conditions have revealed key factors of immunity to EBV, but also more generally of the immune system in humans. However, there are still patients with a high susceptibility to EBV of unknown genetic origin.

**Results**

Two siblings with severe persistent EBV infection leading to EBV-driven Hodgkin lymphoma were identified and found to be carrier of a homozygous deleterious mutation in *RASGRP1*, leading to the lack of RASGRP1 expression. Loss of RASGRP1 impairs T-cell proliferation including CD27- and CTPS1-dependent pathways, two pathways known to be necessary for an efficient immune response to Epstein–Barr virus infection.

**Impact**

Our study highlights the fact that RASGRP1 deficiency is a major risk factor for EBV-driven lymphoproliferative disorders and demonstrates the critical function of RASGRP1 in pathways required for expansion of activated T cells in humans, a key step in immunity to EBV.

dilution 1:1,000), antiphosphorylated ERK 1/2 (#4376S; dilution 1:1,000), anti-ERK 1/2 (#4695S; dilution 1:1,000), antiphosphorylated P38 (#4511S; dilution 1:1,000), antiphosphorylated AKT (serine 473, 4058S; dilution 1:1,000), and anti-Ku70 (4103S; dilution 1:1,000) purchased from Cell Signaling Technology; anti-CTPS1 (EPR8086B; dilution 1:1,000) purchased from Abcam; and anti-actin (A2066; dilution 1:1,000) purchased from Thermo Fischer Scientific; anti-RASGRP1 antibodies from Merck Millipore (MABS146, RASGRP1 epitope recognized unknown; dilution 1:1,000), from Thermo Fischer Scientific (PA5-25750, raised against 495–521 amino acids of RASGRP1; dilution 1:1,000), and from Abcam (EPR9609, raised against the N-terminal part of RASGRP1; dilution 1:1,000). Membranes were then washed and incubated with anti-mouse or anti-rabbit HRP-conjugated antibodies from Cell Signaling (dilution 1:10,000). Pierce ECL Western blotting substrate was used for detection.

### Calcium flux analysis

$Ca^{2+}$ responses were assessed as previously described (Martin *et al*, 2014; Hauck *et al*, 2015).

### Plasmids constructs and infections

Full-length cDNA encoding the mutant RASGRP1 was generated by mutagenesis using the Q5 Site-Directed Mutagenesis Kit (NEB). cDNAs coding wild-type and mutant RASGRP1 were verified by sequencing, inserted into an expression vector pcDNA3.1D/V5-His-TOPO and transfected into HEK293T cells using lipofectamine (Invitrogen). cDNAs were then also inserted into a bicistronic lentiviral expression vector encoding the mCherry protein as a reporter (pLVX-eF1a-IRES-mCherry, Clontech). Viral particles for infection were obtained by co-expression of the lentiviral vector containing RASGRP1 with third-

generation lentiviral plasmids containing Gag-Pol, Rev, and the G protein of the vesicular stomatitis virus (VSVG) into HEK293T. Viral supernatants were collected 48 h after transfection, and viral particles were concentrated by centrifugation. Control and patient's cells were infected at day 2 of PHA stimulation and day one of anti-CD3/CD28 beads stimulation with viral particles, and the mCherry expression was determined by flow cytometry. To assess the selective advantage of mCherry expression during long-term expansion, T-cell blasts were restimulated with anti-CD3 antibody or anti-CD3/CD28 beads. For *RASGRP1* gene knockdown, control cells were infected at day 3 of PHA stimulation and day one of anti-CD3/CD28 stimulation with the pLKO.1 lentiviral vector containing a RASGRP1-specific shRNA (Sigma-Aldrich, n°TCRN0000048271 for #1 and n°TCRN0000048271 for #2), a CTPS1-specific shRNA (OpenBiosystems, n°TRCN0000 045350), or a scramble shRNA in which the puromycin resistance gene was replaced by the GFP gene. The percentage of GFP⁺ cells was assessed by flow cytometry.

### Data availability

The sequencing data from this publication have been deposited to the EGA database (https://www.ebi.ac.uk/ega/) and assigned the identifier EGAS00001002753.

**Expanded View** for this article is available online.

### Acknowledgements

We acknowledge the patient, his family, and the healthy donors for cooperation and blood gifts. S. L. is a senior scientist at the Centre National de la Recherche Scientifique-CNRS (France). We thank the Tumor bank of Tenon Hospital (Paris, France) and Amélie Trinquand et Julie Bruneau from the Department of Pathology at Necker-Enfants Malades Hospital (Paris, France) for providing with tissues of P1.2 and for extraction of DNA. Sylvie Fabrega from the Plateform of Viral Vector and Gene Transfer at Necker-Enfants Malades Hospital (Paris, France) for lentiviral productions. S. W. is supported by Institut IMAGINE, and D. B. is supported by the Agence Nationale de la Recherche (ANR-14-CE14-0028-01 (SL)). This work was supported by grants from Ligue Contre le Cancer-Equipe Labellisée (France) (SL), INSERM (France), the Agence Nationale de la Recherche, France (ANR-14-CE14-0028-01 (SL) and ANR-10-IAHU-01 (Imagine Institut)), and the European Research Council (AF) (ERC-2009-AdG_20090506 n°FP7-249816). Exome sequencing was funded by La Fondation Maladies Rares (SL) (France).

### Author contributions

SW designed and performed experiments, and analyzed the data and clinical information. EM, DB, and CL performed experiments and analyzed the data. SB provided critical reagents and performed experiments. CP, GL, and AP identified the patients, and provided and analyzed clinical information. AF and SW participated in the writing of the manuscript. SL wrote the manuscript, interpreted data, and designed and supervised the research.

### Conflict of interest

The authors declare that they have no conflict of interest.

### For more information

http://exac.broadinstitute.org/gene/ENSG00000172575
http://gnomad.broadinstitute.org/gene/ENSG00000172575
http://omim.org/entry/603962?search=rasgrp1&highlight=rasgrp1

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
