## [Review Process File · EMBO Molecular Medicine]

Loss of RASGRP1 in humans impairs T-cell expansion leading to Epstein-Barr virus susceptibility

Sarah Winter, Emmanuel Martin, David Boutboul, Christelle Lenoir, Sabah Boudjemaa, Arnaud Petit, Capucine Picard, Alain Fischer, Guy Leverger and Sylvain Latour

Review timeline:

Submission date:	19 July 2017
Editorial Decision:	22 August 2017
Revision received:	12 October 2017
Editorial Decision:	15 November 2017
Revision received:	30 November 2017
Accepted:	01 December 2017

Editor: Céline Carret

Transaction Report:

1st Editorial Decision

22 August 2017

Thank you for the submission of your manuscript to EMBO Molecular Medicine. We have now heard back from the three referees whom we asked to evaluate your manuscript.

You will see from the comments pasted below that while the referees find the study of interest, they still do have concerns that must be addressed in a major revision of your work. Referees 1 and 2 have minor issues. Referee 3 however finds that the Nature Immunology paper (Dec 2016) takes away the novelty of the data. While referees 1 and 2 feel that this is a strength of the paper confirming the clinical relevance, referee 3 does not but infers that increasing mechanistic understanding would be desirable to improve the study and confers more novel insights. We would like to encourage to follow this line to make the paper overall more compelling.

I look forward to receiving your revised manuscript.

***** Reviewer's comments *****

Referee #1 (Remarks):

In the paper "Loss of RASGRP1 in humans impairs T cell expansion leading to Epstein-Barr virus susceptibility" Winter et al. describe a consanguineous family with two healthy siblings and two siblings suffering from persistent EBV infection, Hodgkin's lymphoma, and pneumonia (disseminated tuberculosis in patient 1 and pneumocystis jiroveci pneumonia in patient 2) caused by a homozygous loss of function mutation in the RASGRP1 gene leading to a frameshift and loss of the functional C-terminus. A single case with a similar, but not identical mutation presenting with recurrent bacterial and viral infections, low grade EBV-associated B cell lymphoma and failure to thrive had been reported a year ago in another consanguineous family. This first family had lost three older siblings up to the age of 2 years.

The authors confirm the previous report that RASGRP1 plays an important role in MAP kinase activation and induction of T cell proliferation in response to T cell receptor activation by anti-CD3 and anti-CD3 plus anti-CD28 stimulation. They show furthermore that CD27-mediated T cell

proliferation is severely impaired in coculture with EBV-infected B blasts expressing CD70 and that induction of CTPS1 (Cytidin triphosphate synthase 1) under these conditions is abolished.

Transduction of the patient's T cells with a lentivirus expressing wild type RASGRP1 restored CD27-mediated T cell proliferation and induction of CTPS1. This is the fourth gene in the pathway of CD27-mediated T cell activation in response to B blasts (CD27, CD70, CTSP1, and RASGRP1) leading to an immunodeficiency that is associated with particular susceptibility to EBV infection. This is very good work at the front edge of immunodeficiency research. The experiments are clearly described and well performed. Given that only a single case of RASGRP1 deficiency has been reported in the literature so far, this is an important contribution to the field. It not only confirms the importance of the RASGRP1 gene product in T cell immunity, it also extends our knowledge of the CD70-CD27 pathway: it underlines its general importance for the immune response and lays special emphasis on its role in T-cell mediated immunity to EBV.

One major and two minor points:

It seems to me as if in the right part of Figure 3A (anti-CD3 and anti-CD28 beads) the graphs in the two upper lanes are mixed up. In the controls (upper lane) the cells are shifted to the left (indicative of proliferation) to a lesser extent than in the second lane (Pat.). The histogram presented in the outmost right graph of the third lane of Figure 3A is contradictory to what is seen in the two upper graphs.

The two minor points:

1. On page 6, lane 4 it should read: In patient cells, global tyrosine phosphorylation of substrates of the TCR signaling cascade was not different from that observed in control cells, albeit Ca⁺⁺ mobilization and basal phosphorylation of PLC- γ 1 were found to be slightly enhanced (Fig 2A and B and upper lane of 2C).
2. The x-axis in the supplementary figure 2A and B is labeled: FCS. I assume this should read: FSC.

Referee #2 (Comments on Novelty/Model System):

The description here of siblings with homozygous mutations in RASGRP1 and the mechanistic work performed by the authors adds to our understanding of EBV immunity and T cell function. Mutations in this gene have been reported once before but this work goes much further. The model is appropriate and technical quality sufficient. Descriptions of phenotypes such as this are important for the medical community and the authors highlight potential therapeutic targets and further work.

Referee #2 (Remarks):

Overall the manuscript is clear but language needs revision in places.

Specific comments:

Please could the authors expand on the patients episodes of pneumonia - was this prior to transplant/chemo? This sentence ("Both patients also...") requires revision to provide clarity for the reader.

Is TRECS data available - if so this should be included

The use of the word 'peculiar' in relation to EBV susceptibility would better be changed to specific
 Could the authors propose a mechanism for the lymphopaenia described in these patients - this should be added before publication - they address possible reasons for absence of NKT cells but do not address low NK and T cell numbers. Can anything be drawn from the mouse model? Are the animals lymphopaenic?

Revision of language in methods section also required

Referee #3 (Remarks):

In the submitted manuscript, Winter et al. describe two patients with persistent EBV infections that eventually led to Hodgkin lymphoma. These patients had homozygous mutations in RasGRP1 resulting in a premature stop and loss of protein expression. The patients share similarities with a RasGRP1-deficient patient described in Nature Immunology last year (Salzer et al, Nature Immunology 2016). The authors here show that loss of RasGRP1 led to defective MAPK signalling, enhanced basal phosphorylation of PLC γ and slightly enhanced Ca²⁺ influx. Cell biologically, patient T cells displayed a proliferation defect, particularly in response to CD70

stimulation which is important for controlling EBV infection. There are many similarities with the more extensive Salzer study, although the authors here also found that patient T cells lacking RasGRP1 have defective upregulation of the enzyme CTPS-1 which has been shown to be protective against EBV infection. Overall, the work presented is solid but does not truly provide new insights or mechanisms. We therefore see this more fit for a specialized (immunological) journal and not for EMBO.

Major point:

The authors claim that the truncation in RasGRP1 suppresses protein but not transcript expression in the patients. The authors draw this conclusion from western blotting using a monoclonal antibody against an unknown epitope of RasGRP1. The authors should verify this result using an antibody known to bind to the N-terminus of RasGRP1, or by epitope tagging RasGRP1 in the construct used for overexpression. If it turns out that this mutant RasGRP1 is expressed, this could provide interesting mechanistic explanation for phenotypic differences compared to Salzer et al.

Other points:

The authors should compare and contrast their results with Fuller et al 2007 where a similar truncation is introduced into a mouse model. Along those lines, the authors should discuss the possibility that loss of RasGRP1 may influence thymic selection.

The authors propose that loss of RasGRP1 leads to defective proliferation in part through loss of CTPS-1 upregulation upon TCR stimulation. This conclusion would be greatly supported by experiments studying effects of CTPS-1 knockdown on proliferation of patient and control cells.

Minor points:

- Ras/RasGRP1 are not written in all caps.
- On page 7, the reference to figure 2A and 2B should refer to 2B and 2C
- Label the y-axes for flow cytometry histograms
- The authors refer several times to "data not shown." Including this data would add to the characterization of the effects of this mutation.

1st Revision - authors' response

12 October 2017

Referee #1 (Remarks):

In the paper "Loss of RASGRP1 in humans impairs T cell expansion leading to Epstein-Barr virus susceptibility" Winter et al. describe a consanguineous family with two healthy siblings and two siblings suffering from persistent EBV infection, Hodgkin's lymphoma, and pneumonia (disseminated tuberculosis in patient 1 and pneumocystis jiroveci pneumonia in patient 2) caused by a homozygous loss of function mutation in the RASGRP1 gene leading to a frameshift and loss of the functional C-terminus. A single case with a similar, but not identical mutation presenting with recurrent bacterial and viral infections, low grade EBV-associated B cell lymphoma and failure to thrive had been reported a year ago in another consanguineous family. This first family had lost three older siblings up to the age of 2 years.

The authors confirm the previous report that RASGRP1 plays an important role in MAP kinase activation and induction of T cell proliferation in response to T cell receptor activation by anti-CD3 and anti-CD3 plus anti-CD28 stimulation. They show furthermore that CD27-mediated T cell proliferation is severely impaired in coculture with EBV-infected B blasts expressing CD70 and that induction of CTPS1 (Cytidin triphosphate synthase 1) under these conditions is abolished. Transduction of the patient's T cells with a lentivirus expressing wild type RASGRP1 restored CD27-mediated T cell proliferation and induction of CTPS1. This is the fourth gene in the pathway of CD27-mediated T cell activation in response to B blasts (CD27, CD70, CTSP1, and RASGRP1) leading to an immunodeficiency that is associated with particular susceptibility to EBV infection. This is very good work at the front edge of immunodeficiency research. The experiments are clearly described and well performed. Given that only a single case of RASGRP1 deficiency has been reported in the literature so far, this is an important contribution to the field. It not only confirms the importance of the RASGRP1 gene product in T cell immunity, it also extends our knowledge of the

CD70-CD27 pathway: it underlines its general importance for the immune response and lays special emphasis on its role in T-cell mediated immunity to EBV.

One major and two minor points:

It seems to me as if in the right part of Figure 3A (anti-CD3 and anti-CD28 beads) the graphs in the two upper lanes are mixed up. In the controls (upper lane) the cells are shifted to the left (indicative of proliferation) to a lesser extent than in the second lane (Pat.). The histogram presented in the outmost right graph of the third lane of Figure 3A is contradictory to what is seen in the two upper graphs.

We thank the referee to have noticed this mistake. We apologize for this. Indeed, the two upper dot plots (nonstimulated (I) and anti-CD3/CD28 beads) have been inverted with the lower panels. Upper dot plots correspond to the patient and the lower panels to the control. This has been corrected in the revised version of the manuscript.

The two minor points:

On page 6, lane 4 it should read: In patient cells, global tyrosine phosphorylation of substrates of the TCR signaling cascade was not different from that observed in control cells, albeit Ca⁺⁺ mobilization and basal phosphorylation of PLC- γ 1 were found to be slightly enhanced (Fig 2A and B and upper lane of 2C).

This has been changed in the text following the sentence of the referee.

2. The x-axis in the supplementary figure 2A and B is labeled: FCS. I assume this should read: FSC.

This has been corrected in the revised version of the manuscript.

Referee #2 (Comments on Novelty/Model System):

The description here of siblings with homozygous mutations in RASGRP1 and the mechanistic work performed by the authors adds to our understanding of EBV immunity and T cell function. Mutations in this gene have been reported once before but this work goes much further. The model is appropriate and technical quality sufficient. Descriptions of phenotypes such as this are important for the medical community and the authors highlight potential therapeutic targets and further work.

Referee #2 (Remarks):

Overall the manuscript is clear but language needs revision in places.

We have improved the language throughout the text in the revised version of the manuscript.

Specific comments:

Please could the authors expand on the patients episodes of pneumonia - was this prior to transplant/chemo?

This sentence ("Both patients also...") requires revision to provide clarity for the reader.

Patients developed episodes of pneumonia while they were not receiving any immunosuppressive treatment. We now clarified this point in the revised version of the manuscript.

Is TRECS data available - if so this should be included

Unfortunately, TRECS data were not available for the two patients

The use of the word 'peculiar' in relation to EBV susceptibility would better be changed to specific

We changed "peculiar" to "specific" throughout the text as suggested by the referee (page 11 and page 6).

Could the authors propose a mechanism for the lymphopaenia described in these patients - this should be added before publication - they address possible reasons for absence of NKT cells but do not address low NK and T cell numbers. Can anything be drawn from the mouse model? Are the animals lymphopaenic?

We have now addressed in the discussion the possible mechanisms of the NK and T-cell lymphopaenia found in the patients based on mice models for RASGRP1 deficiency.

Revision of language in methods section also required

We revised the language of the methods section.

Referee #3 (Remarks):

In the submitted manuscript, Winter et al. describe two patients with persistent EBV infections that eventually led to Hodgkin lymphoma. These patients had homozygous mutations in RasGRP1 resulting in a premature stop and loss of protein expression. The patients share similarities with a RasGRP1-deficient patient described in Nature Immunology last year (Salzer et al, Nature Immunology 2016). The authors here show that loss of RasGRP1 led to defective MAPK signalling, enhanced basal phosphorylation of PLCgamma and slightly enhanced Ca²⁺ influx. Cell biologically, patient T cells displayed a proliferation defect, particularly in response to CD70 stimulation which is important for controlling EBV infection. There are many similarities with the more extensive Salzer study, although the authors here also found that patient T cells lacking RasGRP1 have defective upregulation of the enzyme CTPS-1 which has been shown to be protective against EBV infection. Overall, the work presented is solid but does not truly provide new insights or mechanisms. We therefore see this more fit for a specialized (immunological) journal and not for EMBO.

We disagree with the last comment of the referee. In Salzer et al., the mechanism underlying the EBV susceptibility in RASGRP1 deficiency was not addressed. Our study demonstrates the critical role of RASGRP1 in the expansion of EBV-specific T cells during the immune response to EBV. In particular, we show that RASGRP1 is required for CD70 and CTPS1-dependent proliferation pathways. We have now included new data (in the revised version) suggesting that RASGRP1 is not only required for CTPS1 gene expression, but also for other factors involved in cell proliferation such as PCNA (see the new Extended View Figure 5).

Major point:

The authors claim that the truncation in RasGRP1 suppresses protein but not transcript expression in the patients. The authors draw this conclusion from western blotting using a monoclonal antibody against an unknown epitope of RasGRP1. The authors should verify this result using an antibody known to bind to the N-terminus of RasGRP1, or by epitope tagging RasGRP1 in the construct used for overexpression. If it turns out that this mutant RasGRP1 is expressed, this could provide interesting mechanistic explanation for phenotypic differences compared to Salzer et al.

We now provide new data (in Figure 1 and in Extended View Figure 1) indicating that the mutant protein is not detectable in cells of the patient, while in transient over expression experiments (in HEK293T) a truncated protein can be detected using the same antibody that fails to detect RASGRP1 in the cells of the patient. We also now show that most of RASGRP1 transcripts detected in cells of patient were aberrant out-of-frame transcripts lacking at least exon 16 (see page 6). These data did not change our initial conclusion that this mutation leads to undetectable RASGRP1 protein expression, although we could not exclude residual expression of a truncated product (out of the detection threshold).

Other points:

The authors should compare and contrast their results with Fuller et al 2007 where a similar truncation is introduced into a mouse model. Along those lines, the authors should discuss the possibility that loss of RasGRP1 may influence thymic selection.

We have now compared and discussed the results of Fuller et al. 2007, page 13-14 of the revised version of the manuscript. The possibility that loss of RasGRP1 may also influence thymic selection in humans (like in mice) is also discussed page 13-14.

The authors propose that loss of RasGRP1 leads to defective proliferation in part through loss of CTPS-1 upregulation upon TCR stimulation. This conclusion would be greatly supported by experiments studying effects of CTPS-1 knockdown on proliferation of patient and control cells.

This is shown in Figure 3D, in which knock down of RASGRP1 or CTPS1 in control cells has the same inhibitory effect on proliferation. We did not knock down CTPS1 in RASGRP1-deficient cells

since CTPS1 expression is already strongly decreased.

Minor points:

- *Ras/RasGRP1 are not written in all caps.*

The official nomenclature in human for RasGRP1 is RASGRP1 in uppercases (in lower cases for mice).

- *On page 7, the reference to figure 2A and 2B should refer to 2B and 2C*

We thank the referee to have noticed this mistake. We apologize for this. This has been corrected in the revised version of the manuscript.

- *Label the y-axes for flow cytometry histograms*

We have now labelled the y-axes (normalized to mode).

- *The authors refer several times to "data not shown." Including this data would add to the characterization of the effects of this mutation.*

We have now included most of these data not shown in Figure 4 panel C and in three new additional Extended View Figures (1, 2 and 5).

2nd Editorial Decision

15 November 2017

Thank you for the submission of your revised manuscript to EMBO Molecular Medicine. We have now received the enclosed reports from the referees that were asked to re-assess it. As you will see, while referee 2 remains not satisfied with the amount of mechanistic data provided, the other reviewer is now supportive and I am pleased to inform you that we will be able to accept your manuscript pending the following final amendments:

***** Reviewer's comments *****

Referee #2 (Remarks for Author):

I am happy with the authors' responses to queries and revisions in the manuscript.

Referee #3 (Remarks for Author):

In my opinion there are only limited incremental mechanistic insights provided by this study.

Corresponding Author Name: Sylvain LATOUR
Journal Submitted to: EMBO Molecular Medicine
Manuscript Number: EMM-2017-08292